# Acid-Triggered Switchable Near-Infrared/Shortwave Infrared Absorption and Emission of Indolizine-BODIPY Dyes

**DOI:** 10.3390/molecules28031287

**Published:** 2023-01-29

**Authors:** Matthew A. Saucier, Cameron Smith, Nicholas A. Kruse, Nathan I. Hammer, Jared H. Delcamp

**Affiliations:** 1Department of Chemistry and Biochemistry, University of Mississippi, University, MI 38677, USA; 2Materials and Manufacturing Directorate, Air Force Research Laboratory, 2230 Tenth Street Area B Building 655, Wright-Patterson AFB, Dayton, OH 45433, USA; 3UES, Inc., 4401 Dayton Xenia Rd, Dayton, OH 45432, USA

**Keywords:** near-infrared (NIR) emission, shortwave infrared (SWIR) emission, BODIPY, indolizine, fluorescence imaging agents

## Abstract

Fluorescent organic dyes that absorb and emit in the near-infrared (NIR, 700–1000 nm) and shortwave infrared (SWIR, 1000–1700 nm) regions have the potential to produce noninvasive high-contrast biological images and videos. BODIPY dyes are well known for their high quantum yields in the visible energy region. To tune these chromophores to the NIR region, fused nitrogen-based heterocyclic indolizine donors were added to a BODIPY scaffold. The indolizine BODIPY dyes were synthesized via microwave-assisted Knoevenagel condensation with indolizine aldehydes. The non-protonated dyes showed NIR absorption and emission at longer wavelengths than an aniline benchmark. Protonation of the dyes produced a dramatic 0.35 eV bathochromic shift (230 nm shift from 797 nm to 1027 nm) to give a SWIR absorption and emission (λ_max_^emis^ = 1061 nm). Deprotonation demonstrates that material emission is reversibly switchable between the NIR and SWIR.

## 1. Introduction

The demand for affordable, non-invasive, high-resolution, high-frame-rate biological imaging is obvious [1,2,3]. The growing field of fluorescence imaging has the potential to meet this demand; however, few small-molecule probes provide optimal photophysical and biological properties to be used at the clinical scale [1,2,3]. Optimal photophysical properties include: absorption and emission wavelengths (λ_max_^abs^ and λ_max_^emis^) lying in the near infrared (NIR, 700–1000 nm) or, preferably, in the shortwave infrared (SWIR, 1000–1700 nm) regions, strong molar absorption coefficients (ε), and high emissive quantum yields (*Φ*) [4,5,6,7,8,9]. Longer emission wavelengths offer diagnostic imaging capability with improved signal-to-noise ratios, due to the diminishing autofluorescence and scattering of biological tissue [1,10,11]. The product of molar absorptivity and emissive quantum yield gives molecular brightness (ε × *Φ* = MB). A high MB at longer wavelengths is crucial for minimizing dye dosage and maximizing the imaging signal [10,11]. Designing a molecule with a high MB and SWIR emission would improve the quality, accuracy, and imaging feedback time of current non-invasive in vivo fluorescence imaging.

Boron-dipyrromethene (BODIPY) is a fused nitrogen- and boron-containing heterocyclic fluorophore core that is known for bright emission in the visible and NIR regions [12,13,14,15,16]. When compared to cyanines and xanthenes, few BODIPY dyes emit in the SWIR [17,18,19,20,21]. Tuning electron donating groups on a BODIPY scaffold can potentially produce bright NIR/SWIR dyes. Indolizine is a fused nitrogen-based heterocycle that displays exceptional electron donating ability due to its planarity, full-conjugation, and proaromaticity in the donated configuration [22]. Our group has recently reported squaraines [23,24], cyanines [25,26], and xanthenes [18,27] equipped with indolizine donors to replace common amine and indoline donors, which significantly shifted the parent chromophores to longer emission wavelengths. Thus, we aimed to do the same with a classic BODIPY scaffold to embark on a fundamental chromophore design study similar to our previous publications on cyanines and squaraines [23,25]. These fundamental studies are critical to designing desirable chromophore properties and synthetic routes to these systems before synthetically complicating water solubilizing functionality is installed for biological imaging purposes. Additionally, according to the literature, *meso*-C BODIPY dyes (where a carbon is located at the middle, or *meso*, position in the BODIPY core) seem to rely heavily on extended conjugation, as well as donation strength, to access longer wavelengths. Installing styryl substituents on a tetramethyl BODIPY bathochromically shifts the λ_max_^emis^ by 0.64 eV (186 nm shift from 514 nm to 700 nm; Figure 1) [28,29]. However, exchanging the phenyl substituents (no donors) with dimethylanilines (strong donors) minimally affects the λ_max_^emis^ (700 vs. 737 nm) [12]. Herein, we show that using an indolizine donor in place of the dimethyl aniline donor produces a significant bathochromic shift of the λ_max_^emis^ by 0.26 eV (135 nm shift from 737 nm to 872 nm) in the *meso*-C BODIPY scaffold that is well into the NIR region.

## 2. Results and Discussion

### 2.1. Computational Studies

The neutral indolizine BODIPY dyes (**2Ph** and **1Ph**) and the dimethylaniline literature benchmark dye (**DMA**) were evaluated computationally to predict absorption maxima, HOMO and LUMO orbital localization, and protonation position. These data were determined by density functional theory (DFT) at the B3LYP/6-311G (d,p) level of theory with dichloromethane implicit solvation via a polarizable continuum model [30,31,32,33,34,35]. All neutral targets exhibited NIR (**DMA** = 708 nm, **2Ph** = 765 nm, and **1Ph** = 761 nm) vertical transitions, with the indolizine-based dyes exhibiting lower energy vertical transitions compared to the aniline benchmark. All targets also exhibited high oscillator strengths (>0.86, Appendix A). For the two novel targets, the HOMO and, to a slightly lesser extent, the LUMO orbitals were delocalized across the indolizine donors and BODIPY core, which indicated potential π–π* behavior (Figure 2). To understand how the bulky phenyl substituent on the indolizine affected the photophysical properties of the indolizine BODIPY scaffold, both the 2-phenyl (**2Ph**) and 1-phenyl (**1Ph**) indolizine donors were selected as synthetic targets for this study. The dimethylaniline BODIPY dye (**DMA**) was also prepared to compare the effects of changing an aniline to an indolizine on the BODIPY scaffold [12].

### 2.2. Synthesis

Both starting material indolizines (**2Ph-Indz** [22] and **1Ph-Indz** [36]), the tetramethyl-BODIPY core (**1** [37]), and dimethylaniline BODIPY (**DMA** [12]) were all prepared according to the literature precedent. The indolizines were formylated via the Vilsmeier–Haack reaction with phosphorus(V) oxychloride (POCl_3_) and *N,N*-dimethylformamide (DMF) in 1,2-dichloroethane (DCE) to afford the corresponding indolizine aldehydes in good yield (Figure 1). **2Ph-CHO** had been previously reported [38], and novel **1Ph-CHO** was formed in an 84% yield. Finally, the indolizine aldehydes were reacted with **1** in a microwave-assisted Knoevenagel-type condensation, using conditions modified from literature procedures [39,40], to afford the indolizine-BODIPY products (**2Ph** and **1Ph**) in moderate to low yields (11–46%). Such condensation reactions between **1** and aryl aldehydes have been reported to be low yielding [41,42,43], especially with increased donation into the aldehyde, as was seen with the indolizine aldehydes. The final zwitterionic indolizine BODIPY dyes were isolated via a short silica plug, suspension of the solid dye in hot hexanes with sonication, and centrifugation to collect the pure dye. The dyes were stable under ambient atmosphere in a dark refrigerator for up to one month. A slight decomposition is observable on longer time scales to give the starting material aldehyde, which can be removed with the aforementioned hexanes sonication and centrifugation treatment. Additionally, the dyes exhibited minimal change in their absorption spectra after 5 days of stirring in a dilute dichloroethane solution (room temperature, dark, and ambient atmosphere; Appendix A). Finally, **2Ph** and **1Ph** were characterized via ^1^H, ^19^F, and ^11^B NMR, along with IR spectroscopy and high-resolution mass spectrometry.

### 2.3. Photophysical Studies

The photophysical properties of all three BODIPY dyes were studied in dichloromethane (CH_2_Cl_2_). All three dyes absorbed and emitted in the NIR region (Table 1 and Figure 3a and Appendix A). As predicted computationally, **2Ph** and **1Ph** exhibited bathochromic shifts in absorption (798 nm and 797 nm, respectively) and emission (867 nm and 872 nm, respectively) compared to **DMA** (707 nm absorption and 760 nm emission). **1Ph** emitted at a slightly longer wavelength than **2Ph**. The molar absorptivity of **1Ph** was higher than that of **2Ph** (121,000 M^−1^ cm^−1^ and 97,000 M^−1^ cm^−1^, respectively), most likely owing to less steric hindrance caused by the phenyl substituent at the 2-position on the indolizine in **2Ph**. The emissive quantum yield of **1Ph** was also higher than that of **2Ph** (5.6% and 3.5%, respectively, ref. IR820, *Φ* = 4.4% [44]), which was possibly due to the reduced steric hindrance of the indolizine donor leading to less non-radiative decay. Noticeably, the emissive quantum yield of **DMA** (34.6%, ref. HITCI, *Φ* = 28.3% [45]) was an order of magnitude higher than **2Ph** and **1Ph**. The product of the molar absorptivity and quantum yield of **2Ph** and **1Ph** gave good MB values at 3400 M^−1^ cm^−1^ and 6800 M^−1^ cm^−1^, respectively, for NIR emission wavelengths.

Inspired by the dramatic bathochromic shift in absorption and emission upon the treatment of azulene BODIPY dyes with trifluoroacetic acid (TFA) that was reported by Zhao, et al. [46], **DMA**, **2Ph**, and **1Ph** were subjected to 1% TFA solution in CH_2_Cl_2_ (ca. 10 μM dye concentration, Figure 3b). As expected, at least one of the aniline donors of **DMA** was protonated, which hypsochromically shifted the λ_max_^abs^ to 627 nm with no lower energy peaks observed (Appendix A). Similarly, the λ_max_^abs^ of **2Ph** hypsochromically shifted to 574 nm, which suggested protonation of the indolizine nitrogen, therefore quenching electron donation into the π-system of the BODIPY dye. However, a peak began forming at 981 nm and continued to grow over 24 h as the 574 nm peak decreased in intensity. At 24 h, the 981 nm peak changed shape slightly and reached its maximum intensity. Beyond 24 h, the peak continued to change shape and decrease in intensity (Appendix A). Interestingly, when subjected to the acidic conditions, **1Ph** immediately formed an absorption peak at 1027 nm, which maintained its shape and increased in intensity to become more intense than the original non-protonated dye absorption peak after 2 h. Between 2 h and 21 h, the shape of the peak changed, and its intensity decreased (Appendix A). The protonated dye, **1Ph-TFA** (**1Ph** in CH_2_Cl_2_ with 1% TFA after sitting for 2 h), displayed a high molar absorptivity of 133,500 M^−1^ cm^−1^, more than that of the original dye. **1Ph-TFA** also emitted weakly in the SWIR region, with a λ_max_^emis^ at 1061 nm and a 0.0020% quantum yield (ref. IR-1061, *Φ* = 0.32% [47], Table 1). 

Lastly, the protonation of **1Ph** was found to be reversible (Figure 3c). **1Ph** was dissolved in a DMSO-*d_6_* solution (ca. 3 mM), then 3000 equivalents of TFA-*d* were added and mixed thoroughly. Deuterated solvents were selected in an attempt to monitor the reaction by ^1^H NMR and UV-vis-NIR absorption spectroscopy; however, the NMR study gave poorly resolved peaks. This solution immediately produced the absorption spectrum in Figure 3c, which is titled **1Ph-TFA**. Then, 3000 equivalents of triethylamine (TEA) were added and mixed thoroughly. This solution immediately produced the absorption spectrum in Figure 3c, which is titled **1Ph-TFA-TEA**. The original dye absorption peak at 797 nm was reformed with an enhanced shoulder peak around 700 nm, which was most likely a by-product of the protonation–deprotonation reaction. The solutions of **1Ph**, **1Ph-TFA**, and **1Ph-TFA-TEA** displayed colorimetric changes upon reversible protonation (Figure 3d). This reversible protonation was observed for **DMA** and **2Ph** as well (Appendix A). Interestingly, the protonation of **2Ph** and **1Ph** with 1% AcOH (*v*/*v*) only partially formed the SWIR peak (Appendix A).

Attempts to elucidate the structure of the protonated dye **1Ph-TFA** by ^1^H NMR were complicated by broadened peaks when using TFA-*d* and DMSO-*d_6_*. Thus, computational investigations were undertaken to help identify a plausible protonated structure. To the optimized geometry of **1Ph**, a single proton was added above the electron rich six-membered ring of one indolizine donor, which was equidistant (~2.3 Å) from each of the 6 atoms in the ring. The geometry was then optimized, and data was obtained by DFT at the B3LYP/6-311G (d,p) level of theory with dichloromethane implicit solvation. Interestingly, the proton was added at the 5-position (at a carbon) of the indolizine. This protonated structure (Figure 4a) was consistent with other examples in the literature of SWIR-emitting BODIPY dyes where an additional cation has been introduced in conjugation via synthetic design or protonation [46,48]. **1Ph-TFA** exhibited a 954 nm vertical transition (Appendix A), which was 0.33 eV lower in energy than that of neutral **1Ph**. That computational shift in vertical transition was similar to the experimental shift in absorption (0.35 eV) seen in Figure 3b, which supported the structure proposed in Figure 4a as being a plausible structure responsible for the observed SWIR absorption and emission. The HOMO and LUMO orbitals (Figure 4b) indicated slight charge transfer behavior, with electron density slightly shifting from the neutral indolizine donor to the cationic indolizine. Additionally, the orbitals indicated a fully conjugated π system, which supports the hypothesis that a cation was likely introduced in conjugation without the proton disrupting conjugation in order to produce the bathochromic shift observed.

## 3. Experimental Details

### 3.1. General Experimental and Computational Information

All reagents, solvents, and starting materials were purchased from commercial vendors and were used without further purification. Microwave synthesis was conducted using a CEM Discover 1.0 with 10 mL CEM glass vials and caps. All microwave reaction conditions included irradiation with 300 W of power, a maximum pressure setting of 150 psi, a maximum run time (time to ramp temperature up to reaction temperature) of 5 min, temperature monitoring using a built-in IR sensor, and compressed air cooling after reaction hold time. Thin-layer silica gel chromatography (TLC) was conducted with Sorbent Technologies, Inc. (Atlanta, GA, USA) glass-backed 250 μm Silica Gel XHL TLC plates and visualized under a UV (254 nm) lamp. Flash column chromatography was performed on a Teledyne CombiFlash Rf+ with prepacked Silica Luknova SuperSep 12–80 g cartridges. Short plugs of silica were performed with Sorbtech technical grade 60 Å pore size (230 × 400 mesh) silica gel. In all cases where flash column chromatography was used, a gradient was established from 100% nonpolar solvent that increased to and went beyond the ratio of solvents at which each compound was eluted, which was reported herein. The ^1^H, ^19^F, ^11^B, and ^13^C NMR spectra were recorded on a Bruker Avance-300 (300 MHz) or Bruker Avance-400 (400 MHz) spectrometer, and the chemical shifts were reported in ppm using residual solvent signals as internal standards (chloroform-*d* δ = 7.26 ppm or DMSO-*d_6_* δ = 2.50 ppm for ^1^H NMR, and chloroform-*d* δ = 77.16 ppm for ^13^C NMR). Data were reported as s = singlet, d = doublet, t = triplet, q = quartet, p = pentet, m = multiplet, br = broad; coupling constants, *J*, were in Hz. For electrospray ionization (ESI) high-resolution mass spectrometry (HRMS), quadruple-TOF was used to obtain the data, both in positive and negative modes, with a Waters Synapt HDMS or Orbitrap Exploris 240 to obtain the data in positive mode with a spray voltage of 3600 V, a resolution of 240,000, the ion transfer tube temperature set at 300 °C, and the mass analyzer set to the 200–2000 Da range. ATR-IR was taken using a Bruker Alpha Platinum-ATR FTIR Spectrometer, and spectra were processed on OPUS 6.5 software. All IR samples were taken as a solid/neat, unless otherwise noted. UV-vis-NIR-SWIR absorption spectra were measured using an Avantes/AvaSpecULS2048-USB2-50 spectrometer (Pine Research part RRAVSP3) with an Avantes/AvaSpec light source (Pine Research part RRAVSP) and AvaSoft8 software program, and the Ocean Insight Flame-NIR+ spectrometer (FLMN02855) with an Ocean Insight Halogen light source (HL-2000) and Ocean Insight OceanView software program were used.

The emission data for **DMA** were collected using a Horiba FluoroMax SpectroFluorimeter with a 685 nm excitation source, an excitation slit width resolution of 2 nm, and an emission slit width resolution of 2 nm, with HITCI in EtOH as the reference dye (*Φ* = 23.8% [45]). The emission data of **2Ph** and **1Ph** were collected using a Horiba LabRAM HR Evolution Raman spectrometer with a 785 nm excitation laser with a hole size of 100 μm and a 600 g/mm. The detector used was a silicon-based CCD detector with IR820 in MeOH as the reference dye (*Φ* = 4.4% [44]). The emission data for **1Ph-TFA** were collected using a Horiba PTI QuantaMaster QM-8075-21 fluorometer with a liquid nitrogen cooled InGaAs detector that used a 992 nm excitation source, an excitation slit width of 10 mm, and an emission slit width of 15 mm, with IR-1061 in CH_2_Cl_2_ as the reference dye (*Φ* = 0.32% [47]). Rectangular 10 mm path cuvettes were used for all fluorescence measurements under ambient atmosphere. The photoluminescent quantum yields (PLQY) of the dyes were determined using the integrated emission intensity values (summation of all Y-value data points using Microsoft Excel) by using the relative quantum yield equation [49]:$$\Phi_{sample}=\Phi_{reference}\times \frac{E_{sample}}{E_{reference}}\times \frac{S_{reference}}{S_{sample}}\times \frac{\eta^{2}{}_{sample}}{\eta^{2}{}_{reference}}$$
where *Φ* denotes the quantum yield; *E* refers to integrated emission intensity; *S* is equal to 1–10^−A^, with the superscript A being the absorbance value at the excitation wavelength; *η* is the refractive index of the solvent; sample is the dye studied herein; and reference is the reference standard chosen for quantum yield studies. The absorbances were 0.10–0.33 at the excitation wavelength for all of the dye emission samples. Quantum yields were calculated as an average of multiple emission experiments (three experiments for **2Ph** and **1Ph**, two experiments for **DMA** and **1Ph-TFA**). The error reported was calculated as the standard deviation between the independent quantum yield calculations. Molar absorptivity values were determined using the Beer–Lambert law equation with three individual concentrations plotted with a linear fit (R^2^ >0.99) whose slope gave the molar absorptivity and was rounded to the nearest 500 M^−^^1^ cm^−^^1^.

For all computations, molecules were drawn in ChemDraw (18.0.0.231) and saved as an MDL Molfile. These geometries were then optimized with the MMFF94 force field via Avogadro (1.2.0). Accurate geometry optimization was performed sequentially by DFT using Gaussian 16 [50] with the B3LYP functional [30,31] with the following basis sets: first 3-21G, second 6-31G (d,p) [51,52], and finally 6-311G (d,p) [32]. These were all in dichloromethane as a polarizable continuum model (PCM) [33,34,35]. Time-dependent density functional theory (TD DFT) computations were performed with optimized geometries with the B3LYP functional and 6-311G (d,p) basis set to compute vertical transition energies and oscillator strengths. All optimized geometries displayed no negative vibrational frequencies, which were computed with optimized geometries with the B3LYP functional and 6-311G (d,p) basis set.

### 3.2. Procedure for the Preparation of 1-Phenylindolizine-3-Carbaldehyde (**1Ph-CHO**)

To a 25 mL flame-dried round bottom flask under nitrogen atmosphere charged with a magnetic stir bar at 0 °C (ice bath), anhydrous 1,2-dichloroethane (DCE, 0.4 M concentration with respect to limiting reagent, 7.2 mL), phosphorous(V) oxychloride (POCl_3_, 1.2 equiv., 5.742 mmol, 880 mg, 0.535 mL), and dimethylformamide (DMF, 1.5 equiv., 7.178 mmol, 525 mg, 0.556 mL) were added and stirred for 20 min. at 0 °C. Then, the starting material 1-phenylindolizine (**1Ph-Indz**, 1.0 equiv., 4.785 mmol, 924 mg) was added as a solution in DCE (remaining amount, 4.8 mL) at 0 °C. The resulting reaction mixture was stirred while warming up to room temperature until all the starting material indolizine was consumed according to TLC (4 h). The reaction was quenched with KOH_(aq)_ and extracted with CH_2_Cl_2_. The organic layer was dried with anhydrous Na_2_SO_4_, and the solvent was removed in vacuo. The crude was then purified with flash column chromatography using 15% EtOAc/85% hexanes to afford the pure aldehyde product **1Ph-CHO** in 84% yield (4.038 mmol, 894 mg) as a brown solid (0.25 R_f_ in 10% EtOAc/90% hexanes). The ^1^H NMR (400 MHz, chloroform-*d*) shifts were δ 9.77 (s, 1H), 9.75 (d, *J* = 7.3 Hz, 1H), 7.87 (d, *J* = 9.0 Hz, 1H), 7.58 (d, *J* = 7.6 Hz, 2H), 7.55 (s, 1H), 7.48 (t, *J* = 7.5 Hz, 2H), 7.34 (t, *J* = 7.4 Hz, 1H), 7.30–7.18 (m, 1H), and 6.97 (t, *J* = 6.9 Hz, 1H) ppm. The ^13^C NMR (101 MHz, chloroform-*d*) shifts were δ 177.6, 136.9, 134.6, 129.1, 128.9, 128.0, 126.9, 125.8, 125.4, 123.8, 118.8, 117.9, and 114.6 ppm. The IR (neat, cm^−1^) peaks were 1638, 2776, 2807, 2836, 3000, 3031, 3060, 3084, and 3109. The HRMS (ESI) in *m*/*z* calculated for C_15_H_12_NO [M + H]^+^ was 222.092, and 222.091 was found. The starting material indolizine **1Ph-Indz** [36] and indolizine aldehyde **2Ph-CHO** [38] were synthesized according to the literature precedent.

### 3.3. General Procedure for the Preparation of Indolizine-BODIPY Dyes **2Ph** and **1Ph**

To a 10 mL flame-dried CEM microwave vial under nitrogen atmosphere charged with a magnetic stir bar and ~300 mg oven-dried 3Å molecular sieves, methyl 4-(4,4-difluoro-1,3,5,7-tetramethyl-3a,4a-diaza-4-bora-s-indacen-8-yl) benzoate (**1**, 1.0 equiv.), indolizine aldehyde (4.0 equiv.), anhydrous toluene (0.035 M concentration with respect to limiting reagent), and piperidine (0.265 M concentration with respect to limiting reagent) were added. The microwave vial was then stirred and subjected to microwave irradiation at 150 °C for 1-h increments until all the starting material BODIPY (**1**) and mono-substituted intermediate was consumed by TLC and UV-vis-NIR absorption of the reaction mixture samples diluted in CH_2_Cl_2_. The reaction mixture was then quenched with water & extracted multiple times with CH_2_Cl_2_ until the aqueous layer was colorless. The organic layer was dried with anhydrous Na_2_SO_4_, and the solvent was removed in vacuo. The crude mixture was then passed through a plug of silica gel with a mixture of CH_2_Cl_2_/hexanes (70/30 for **2Ph**, 80/20 for **1Ph**). The solvent was then removed in vacuo. The resulting silica gel purified mixture was suspended in hot hexanes and sonicated, then centrifuged for >1 h, at which point the supernatant was pipetted off. This sonication/centrifugation procedure was repeated multiple times until the black powdery pellet became pure dye by NMR. The starting material BODIPY **1** [37] was synthesized according to the literature precedent. Note that these indolizine BODIPY dyes were not soluble enough in DMSO-*d_6_* to obtain an adequate ^13^C NMR. Dimethylaniline BODIPY **DMA** was prepared under these modified conditions, and its spectra matched those published in the literature [12].

*Methyl 4-(5,5-difluoro-1,9-dimethyl-3,7-bis((E)-2-(1-methyl-2-phenylindolizin-3-yl)vinyl)-5H-4λ4,5λ4-dipyrrolo[1,2-c:2′,1′-f]*[1–3]*diazaborinin-10-yl) benzoate* (**2Ph**): Together, **1** (1.0 equiv., 0.0523 mmol, 20 mg), **2Ph-CHO** (4.0 equiv., 0.2093 mmol, 49 mg,), piperidine (0.265 M, 0.20 mL), and anhydrous toluene (0.035 M, 1.5 mL) were used to afford **2Ph** in an 11% yield (0.0059 mmol, 5 mg) as a black/red solid (0.50 R_f_ in 60% CH_2_Cl_2_/40% hexanes) after reacting for 5 h with a plug of silica with 70% CH_2_Cl_2_/30% hexanes; then, the hexanes sonication/centrifugation procedure was performed as described above. The ^1^H NMR (300 MHz, DMSO-*d_6_*) shifts were δ 8.67 (d, *J* = 6.9 Hz, 2H), 8.10 (d, *J* = 8.0 Hz, 2H), 7.68 (d, *J* = 9.0 Hz, 2H), 7.62–7.46 (m, 8H), 7.41 (d, *J* = 7.4 Hz, 4H), 7.31 (t, *J* = 7.3 Hz, 2H), 7.15–6.93 (m, 6H), 6.79 (s, 2H), 3.90 (s, 3H), 2.23 (s, 6H), and 1.31 (s, 6H) ppm. The ^19^F NMR (376 MHz, DMSO-*d_6_*) shifts were δ −132.8–(−141.1) (br m) ppm. The ^11^B NMR (128 MHz, DMSO-*d_6_*) shifts were δ 0.76 (t, *J* = 34.8 Hz) ppm. The IR (neat, cm^−1^) peaks were 1732, 2861, 2917, 2947, 3028, 3052, 3070, 3094, and 3106. The HRMS (ESI) in *m*/*z* calculated for C_53_H_43_BF_2_N_4_O_2_ [M]^+^ was 816.345, and 816.342 was found.

*Methyl 4-(5,5-difluoro-1,9-dimethyl-3,7-bis((E)-2-(1-phenylindolizin-3-yl)vinyl)-5H-4λ4,5λ4-dipyrrolo[1,2-c:2′,1′-f]*[1,3,2]*diazaborinin-10-yl) benzoate* (**1Ph**): Together, **1** (1.0 equiv., 0.1047 mmol, 40 mg), **1Ph-CHO** (4.0 equiv., 0.4186 mmol, 93 mg,), piperidine (0.265 M, 0.40 mL), and anhydrous toluene (0.035 M, 3.0 mL) were used to afford **1Ph** in a 46% yield (0.0496 mmol, 39 mg) as a black/green solid (0.50 R_f_ in 70% CH_2_Cl_2_/30% hexanes) after using a plug of silica with 80% CH_2_Cl_2_/20% hexanes, and the hexanes sonication/centrifugation procedure was performed as described above. The ^1^H NMR (400 MHz, DMSO-*d_6_*) shifts were δ 8.90 (d, *J* = 7.3 Hz, 2H), 8.15 (d, *J* = 8.2 Hz, 2H), 8.02 (d, *J* = 15.7 Hz, 2H), 7.85 (d, *J* = 9.0 Hz, 2H), 7.74 (d, *J* = 7.3 Hz, 4H), 7.64 (d, *J* = 10.2 Hz, 4H), 7.54–7.42 (m, 6H), 7.33 (t, *J* = 7.4 Hz, 2H), 7.21 (s, 2H), 7.06 (dd, *J* = 8.8, 6.6 Hz, 2H), 6.92 (t, *J* = 6.9 Hz, 2H), 3.93 (s, 3H), and 1.42 (s, 6H) ppm. The ^19^F NMR (282 MHz, DMSO-*d_6_*) shifts were δ −136.1–(−137.0) (br m) ppm. ^11^B NMR (128 MHz, DMSO-*d_6_*) shifts were δ 1.27 (t, *J* = 35.8 Hz) ppm. The IR (neat, cm^−1^) peaks were 1726, 2858, 2918, 2950, 3039, 3075, and 3110. The HRMS (ESI) in *m*/*z* calculated for C_51_H_39_BF_2_N_4_O_2_ [M]^+^ was 788.3143, and 788.3172 was found.

## 4. Conclusions

Two indolizine BODIPY dyes (**2Ph** and **1Ph**) with varying substitution around the indolizine donors were synthesized to compare against the literature benchmark dimethylaniline BODIPY (**DMA**). Installing the indolizine donors in place of the dimethylaniline donor produced a 0.26 eV bathochromic shift (135 nm shift from 737 nm to 872 nm) in the emission peak wavelength that was deep into the NIR region. Exchanging the strong donor from the literature with indolizine donors provided new access to NIR absorbing and emitting BODIPY dyes and expanded the synthetic versatility of the strong indolizine donor in fluorescent dye synthesis. Protonation of the indolizine BODIPY dyes with trifluoroacetic acid showed a dramatic 0.35 eV shift (230 nm shift from 797 nm to 1027 nm) of the maximum absorption peak into the SWIR region, with a maximum SWIR emission at 1061 nm. Protonation of the indolizine BODIPY dyes was shown to be reversible with triethylamine, thus reforming the original dye maximum absorption peaks. This protonation behavior is unique to the indolizine-based dyes, since the maximum absorption peak of **DMA** shifts to shorter wavelengths upon acid exposure. Computational studies support the protonation of an indolizine carbon to place a cation in conjugation with the π-system to lead to a lower energy absorbing material. This switchable NIR/SWIR absorption and emission remains as an interest for further studies to isolate and identify the SWIR fluorophore and expand its mechanism to other dye scaffolds. Since these BODIPY dyes are insoluble in aqueous media, our future directions also focus on making synthetic modifications—similar to those in our previous publication on cyanines and squaraines [24]—to incorporate water-solubilizing groups on the indolizine BODIPY scaffold to allow for aqueous biological imaging.

## Data Availability

Data generated in this study is stored at the University of Mississippi and is available upon reasonable request.

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
