# Peer review of "Acid-Triggered Switchable Near-Infrared/Shortwave Infrared Absorption and Emission of Indolizine-BODIPY Dyes"

_molecules, 2023, doi:10.3390/molecules28031287_

Round 1

Reviewer 1 Report

Delcamp and coworkers report two NIR emissive styryl-BODIPYs conjugated with indolizine donors. They also studied the protonation effects on the styryl-BODIPY conjugates, and they found that upon protonation, the emission maxima shifted to the shortwave infrared (SWIR, 10001700 nm) regions which might be useful for non-invasive in vivo fluorescence imaging applications. The paper is suitable for Molecules as it presents new and interesting findings on this class of compounds. However, a number of issues should be considered before publication:

1.      The11B NMR and the MS spectra are missing from the ESI. Since the NMR spectra indicate the presence of some impurities, they do not provide conclusive data that speak to the purity of their compounds. Given the observed complex photophysics, it is critical that the authors establish the purity of the materials so as to exclude any potential influence of impurities on the observed photophysics.

2.      The authors did not provide any explanation of the bathochromic shift in absorption and emission for the compounds upon treatment with TFA. From ref. 44, it was found that a stable aromatic six-π-electron tropylium cation was formed upon protonation, which was responsible for the bathochromic shift. Please add the explanation for the bathochromic shift, and it would be nice if the authors could provide the HOMO and LUMO orbitals for the protonation products.  

3. The Figure's quality should be improved: Figure 1, particularly, is of poor quality. 

Author Response

Delcamp and coworkers report two NIR emissive styryl-BODIPYs conjugated with indolizine donors. They also studied the protonation effects on the styryl-BODIPY conjugates, and they found that upon protonation, the emission maxima shifted to the shortwave infrared (SWIR, 1000–1700 nm) regions which might be useful for non-invasive in vivo fluorescence imaging applications. The paper is suitable for Molecules as it presents new and interesting findings on this class of compounds. However, a number of issues should be considered before publication:

  1. The11B NMR and the MS spectra are missing from the ESI. Since the NMR spectra indicate the presence of some impurities, they do not provide conclusive data that speak to the purity of their compounds. Given the observed complex photophysics, it is critical that the authors establish the purity of the materials so as to exclude any potential influence of impurities on the observed photophysics.

-MS and 11B NMR spectra are included now. The 11B NMR shows only one signal.

  1. The authors did not provide any explanation of the bathochromic shift in absorption and emission for the compounds upon treatment with TFA. From ref. 44, it was found that a stable aromatic six-π-electron tropylium cation was formed upon protonation, which was responsible for the bathochromic shift. Please add the explanation for the bathochromic shift, and it would be nice if the authors could provide the HOMO and LUMO orbitals for the protonation products.  

-Further computational investigations via DFT have been done and incorporated into the manuscript to identify the protonated product. The computations suggest the six member ring of the indolizine is protonated. The structure is plausible since the protonated dye vertical transition shift was similar to the experimental absorption shift (0.33 eV vs. 0.35 eV, respectively). HOMO and LUMO orbitals of that protonated dye are now included along with a simple structural drawing.

  1. The Figure's quality should be improved: Figure 1, particularly, is of poor quality. 

-We suspect there must have been some type of conversion issue when uploading the files. The figures appear crisp on our drafts. Never the less, we have replaced them with fresh copies of the figures in hopes this solves the problem.

Reviewer 2 Report

In the manuscript, the authors developed two indolizine BODIPY dyes with acid-triggered switchable absorption and emission wavelengths. Although the maximum emission wavelength of the dye lies in NIR or SWIR region, the authors only studied the properties of the dyes in CH2Cl2. Therefore, the performance of the proposed dyes in aqueous environment, which is necessary for biological imaging, was unknown. Moreover, the results provided were inadequate. For example, the 13C NMR of 1Ph and 2Ph, the absorption changes of DMA, 2Ph and 1Ph between 20 h and 80 h, error bar for absorption in Fig. S7 and so on. In addition, grammatical mistakes were found here and there in the manuscript. In view of the above points, I think the manuscript is not suitable for publication in Molecules. 

Author Response

In the manuscript, the authors developed two indolizine BODIPY dyes with acid-triggered switchable absorption and emission wavelengths. Although the maximum emission wavelength of the dye lies in NIR or SWIR region, the authors only studied the properties of the dyes in CH2Cl2. Therefore, the performance of the proposed dyes in aqueous environment, which is necessary for biological imaging, was unknown. Moreover, the results provided were inadequate. For example, the 13C NMR of 1Ph and 2Ph, the absorption changes of DMA, 2Ph and 1Ph between 20 h and 80 h, error bar for absorption in Fig. S7 and so on. In addition, grammatical mistakes were found here and there in the manuscript. In view of the above points, I think the manuscript is not suitable for publication in Molecules. 

- This study is a fundamental science study aimed at designing a chromophore core with photophysical properties in an underexplored spectral region. We are currently pursuing dyes designs centered around this chromophore that have water solubility. As experience in this area has shown our team, it is often not trivial to isolate high purity water soluble dyes without considerable synthetic optimizations beyond the scope of this paper. The focus of this manuscript is in identifying a reasonable chromophore (in terms of stability, synthesis, and quantum yield) that emits in the >800 nm region to then use in biological systems after peripheral modifications. We have had considerable success in implementing this strategy with squaraine and cyanine chromophores with some of these now being texted for crime scene investigations for blood sensing. These materials were also investigated at a fundamental photophysical level first with a long term application in mind much like the materials in this study. We envision these findings launching a series of studies aimed at more applied scenarios bases off of this BODIPY chromophore design.

- 13C NMR for the two dyes were not obtained due to solubility limitations. This is noted in the experimental section and is common to many NIR/SWIR dye designs. The dyes are soluble in CDCl3; however, the 1H NMR spectra show no aromatic peaks most likely due to aggregation. DMSO was able to break up aggregation to produce signals in the 1H NMR which corresponded to the dye but the dye is weakly soluble in this solvent. 13C NMRs obtained with a saturated sample at 50ËšC for 60,000 scans (~70 hours) produced inadequate spectra.

-Additional quantum yield experiments were done and standard deviation errors for the quantum yields have been calculated and now are reported. Extra information has also been provided to show the rigor in our molar absorptivity measurements (see experimental details).

Reviewer 3 Report

The authors present the synthesis and photophysical studies of two indolizine-BODIPY dyes with emission in the NIR region, which can be further shifted to the SWIR upon protonation.

This work is of interest to those involved in the development and application of imaging probes.

Publication is recommended, after the following corrections are made:

-       Too many keywords, please edit.

-       In page 5, lines 144-148, the authors refer the use of deuterated solvents for obtaining absorption spectra, which does not make sense, and in disagreement with the information in Figure 3. Please correct. Also in line 145, “equivalents” rather than “equivalences”.

-       In page 6, lines 209-210, the authors state “A is equal to 1−10−A”. What is this? Please correct, A is the absorbance at the excitation wavelength.

-       In page 6, line 212, working with absorbances as high as 0.5 can raise issues regarding the inner filter effect. Especially, given the very high molar absorptivities and the overlap of the absorption and emission curves. Can the authors comment on this?

-       In page 7, line 230, what is the meaning of “DCE, 0.40M”? Did the authors use neat DCE? If so, why the molarity? The same comment in page 8, lines 273-274 for piperidine and toluene.

-       In page 8, line 299, “shift to longer wavelength) bathochromic shift”. Redundancy, please correct.

Author Response

The authors present the synthesis and photophysical studies of two indolizine-BODIPY dyes with emission in the NIR region, which can be further shifted to the SWIR upon protonation.

This work is of interest to those involved in the development and application of imaging probes.

Publication is recommended, after the following corrections are made:

-       Too many keywords, please edit.

+ Fixed.

-       In page 5, lines 144-148, the authors refer the use of deuterated solvents for obtaining absorption spectra, which does not make sense, and in disagreement with the information in Figure 3. Please correct. Also in line 145, “equivalents” rather than “equivalences”.

+ The figure has been updated.

The experiment was done in deuterated DMSO and with TFA-d while trying to produce an NMR of the protonated dye however the peaks generated were broad and inconclusive in terms of structural identification. However, the same solution was easily tracked by absorption spectroscopy.

“Equivalences” has been corrected.

-       In page 6, lines 209-210, the authors state “A is equal to 1−10−A”. What is this? Please correct, A is the absorbance at the excitation wavelength.

+The relative quantum yield equation uses this formula (1–10-A) as the absorption component of the equation. We agree that it generate confusion, but it is standard for this equation. In order to clarify this, I’ve switched the formula variable “A” to “S” so that there is only one “A” variable which is the absorbance at the excitation wavelength. Maybe some new formate like this can slowly be adopted to avoid confusion.

-       In page 6, line 212, working with absorbances as high as 0.5 can raise issues regarding the inner filter effect. Especially, given the very high molar absorptivities and the overlap of the absorption and emission curves. Can the authors comment on this?

+The high absorbing sample was remeasured with a lower absorption value which did increase the quantum yield value from 30.6% to 34.6%, so we thank the reviewer for catching this as this has revealed a more accurate higher quantum yield number for 1Ph.

+That makes our range now from 0.10-0.33. Our departmental instruments have a limit of detection with dilute samples (especially error associated with very dilute samples on our absorption spectrometer). The new range of values ought to minimize inner filter effects to the best of our abilities, while also maintaining a reasonable level of accuracy in absorption value measurement on our spectrometer.

-       In page 7, line 230, what is the meaning of “DCE, 0.40M”? Did the authors use neat DCE? If so, why the molarity? The same comment in page 8, lines 273-274 for piperidine and toluene.

+ We've updated this. This molarity is with respect to the limiting reagent of the reaction, which helps to standardize reaction concentrations from batch to batch and at any scale.

-       In page 8, line 299, “shift to longer wavelength) bathochromic shift”. Redundancy, please correct.

+ Updated. In cases with nm shifts, the region is now reported to be more accurate.

Round 2

Reviewer 1 Report

The authors have made several modifications accordingly to reviewer comments. The research presented has dramatically improved and has been executed well. In addition, the manuscript's contents are potential of interest to the broad readership of Molecules. Hence the manuscript is recommended for publication.

Author Response

Thank you for reviewing our manuscript and offering suggestions!

Reviewer 2 Report

It is better to explain about the weak water-solubility of the dyes and the further modification to improve it in the manuscript. In addition, is it possible to do solid 13C NMR?

Author Response

Thank you for the suggestion. We've added in a couple of comments to the manuscript in the introduction and SI explaining the direction this work is going as fundamental study and that water solubility enhancing groups are needed as a future study similar to our reply in the prior round of revisions. Our apologies for not adding these comments during the last round of revisions. 

Unfortunately, we do not have solid state 13C capabilities on our campus and can not readily conduct these studies.